# Analysis of the Spatial-Temporal Evolution of Urbanization Quality in Zhejiang Province, China

**DOI:** 10.3390/ijerph20054093

**Published:** 2023-02-24

**Authors:** Yangfei Huang, Xiaomin Jiang, Yong Chen

**Affiliations:** 1School of Civil Engineering and Architecture, Zhejiang University of Science and Technology, 318 Liuhe Road, Hangzhou 310023, China; 2Center of Urban and Rural Development, Zhejiang University of Science and Technology, 318 Liuhe Road, Hangzhou 310023, China

**Keywords:** urbanization quality, space-time evolution, entropy weight method, regional differences, gray correlation calculation method

## Abstract

Taking cities in Zhejiang Province of China from 2011 to 2020 as the research object, a multi-dimensional urbanization quality evaluation index system was constructed using the comprehensive analysis method, and the urbanization quality of 11 cities in Zhejiang Province was quantitatively measured using the entropy weight method. The system classification and time-space evolution analysis were carried out using ArcGIS software (Environmental Systems Research Institute, Inc., RedLands, CA, USA) to comprehensively study the evolution characteristics and influencing factors of the urbanization quality of cities in Zhejiang Province. This study provides a reference for local governments to formulate feasible urbanization development strategies and policies to promote the high-quality development of urbanization and for the construction of new urbanization in other provinces and cities.

## 1. Introduction

The high-quality development of urbanization is the only way to realize modernization, and it is a powerful engine to promote the high-quality development of the national economy of China in the new era [1]. Since the reform and opening up, China has experienced the largest and fastest urbanization process in the history of the world, and the country has made remarkable achievements. The urbanization rate of the permanent population has increased from 17.9% in 1978 to 64.72% in 2021. Cities and towns have become the main carrier of the population and of high-quality development. In the “Fourteenth Five-Year Plan” period, China has entered a new stage of development and the construction of new urbanization has also embarked on a new journey. Improving the quality of urbanization development has become the goal of China’s new urbanization development [2], ensuring that more people enjoy a higher quality of urban life and providing strong impetus and solid support for the basic realization of modernization in 2035.

In recent years, the high-quality development of urbanization has received unprecedented attention from the academic community, primarily focusing on the connotation [3,4,5,6], evaluation system, comprehensive measurement, impact factors, and improvement measures of urbanization quality. For the empirical measurement of urbanization quality, researchers have adopted different dimensions to explore the measurement indicators. For example, Heshmati, A. and Rashidghalam, M. studied urbanization using composite indices of urban infrastructure [7], Fang Chuanglin and Wang Deli formed a three-dimensional indicator ball of urbanization development quality measurement using 12 indicators for the three aspects of economy, society, and space [8], and Liao Haiyan constructed a new urbanization evaluation indicator system in the developed regions of China based on the four aspects of economic development, social equity, environmental optimization, and the quality of people’s livelihoods [9]. Gan Jing et al. constructed a comprehensive measurement index system using 16 indicators in the four dimensions of population, economy, society, and space [10]. Shang Xueying et al. constructed an evaluation system for the urbanization quality in Gansu Province using 18 indicators in the four dimensions of economic development, resident lives, urban and rural coordination, and sustainable development quality [11]. Zhao Yu et al. constructed an indicator system of high-quality urbanization development based on the four dimensions of economic development, scientific and technological support, ecological protection, and coordination and sharing [12]. Li et al. constructed an evaluation system of the urbanization level using the five aspects of population, land, economy, society, and ecology [13]. In terms of measurement methods, the most mature methods include the entropy method [10,14,15,16], principal component analysis method [17,18,19], analytic hierarchy process [20,21], and the data envelopment method [22], which are evaluated from a static perspective. In terms of measurement scale, the research on urbanization quality covers different spatial levels, such as the Belt and Road [23], the country [24,25,26,27], the province [18,28], the trans-regional urban agglomeration [29,30], the city [31,32], and the county [17,33]. In terms of the spatial differentiation and influencing factors of urbanization, Yang Lulu explored the spatial pattern differentiation and evolution of the comprehensive development quality of urbanization in the six central provinces [34]. Wang Yirui et al. analyzed the reasons for the spatiotemporal evolution of China’s urbanization quality and suggested that China’s urbanization quality zoning had a significant ladder [35]. Zhao Yu and other researchers suggested that industrial structure, government guidance, and the utilization of foreign capital were the main factors that led to the spatial differentiation of high-quality urbanization development in the eastern, central, and western regions [12]. Jiang Zhengyun suggested that high-quality urbanization areas would continue to expand along the “T” track, with coastal and riverside areas as the primary axis; the heterogeneity of inter-provincial development would first diffuse and then converge; the quality of new urbanization between provinces had a high degree of spatial dependence; and that lagging economic growth had become the main obstacle to improving the quality of new urbanization in most provinces and regions at the time of the study [36]. Yuan Xiaoling et al. suggested that the social development and urban-rural relations of a region would be affected by the economic level, historical conditions, and geographical factors and that the process of new urbanization between regions had obvious spatial differences and spatial dependence [24]. Luo Tengfei and Deng Hongbing measured the urbanization quality of the Yangtze River Economic Belt and analyzed the spatial and temporal differences. They found that the urbanization development quality of the Yangtze River Economic Belt had a large spatial difference and that there was a relatively significant spatial polarization phenomenon. Factors such as administrative location, resource endowment, and industrial structure affect the spatial distribution of urbanization quality [37]. Based on the existing research, it can be said that there is no recognized indicator system for measuring urbanization quality at present. The research on the space-time evolution of urbanization quality development primarily focuses on the national, regional, or partial provincial levels, while there has been less research conducted on the space-time evolution of high-quality urbanization development in developed provinces.

Zhejiang Province, located in the Yangtze River Delta, borders the East China Sea in the east, Fujian in the south, Anhui and Jiangxi in the west, and Shanghai, Jiangsu, and Anhui in the north, with a total area of 105,500 km^2^ and 11 prefecture-level administrative regions under its jurisdiction. Zhejiang has taken the lead in implementing the new urbanization strategy in China, and its urbanization level is at the forefront of the country. In 2021, the urbanization rate of the permanent population in Zhejiang Province reached 72.7%, and the province entered the mature development period of overall urbanization. The “Fourteenth Five-Year Plan” for the development of new urbanization in the province clearly puts forward the comprehensive implementation of the new urbanization strategy with human-centered high-quality orientation and modernization orientation. According to statistics, the income difference between urban and rural residents in Zhejiang is 1.96, far lower than the national income difference value of 2.56. The incomes of urban and rural residents in Zhejiang have been the highest in the country for 21 years and 37 years, respectively (except for municipalities directly under the Central Government). In June 2022, Zhejiang Province put forward the “two priorities”: to strive to advance the common prosperity of socialism with Chinese characteristics and the provincial modernization in high-quality development, and to continue to implement the new urbanization strategy that is people-oriented, high-quality–oriented, and modernization-oriented. By 2035, Zhejiang will essentially achieve high-level modernization and take the lead in realizing human modernization, urban modernization, industrial modernization, governance modernization, and urban-rural integration. The urbanization rate of the permanent population in the province has reached about 80%, the income ratio of urban and rural residents has been reduced to about 1.60:1, and a national demonstration province of new urbanization has been fully built. However, according to the standards and requirements of the leading province of socialist modernization, imbalance and inadequacy in the process of urbanization in Zhejiang Province still exist, and theoretical research based on the quality development of urbanization in Zhejiang Province is relatively lacking. Based on this, cities in Zhejiang Province from 2011 to 2020 were taken as the research object in this study, and the comprehensive analysis method was used to build a multi-dimensional urbanization quality evaluation index system. The entropy weight method was used to quantitatively measure the urbanization quality of 11 cities in Zhejiang Province. In addition, ArcGIS software was used to carry out systematic classification and spatial-temporal evolution analysis to comprehensively study the evolution characteristics and influencing factors of the urbanization quality of all cities in Zhejiang Province to provide a reference for local governments to formulate practical new urbanization development strategies and policies to promote the high-quality development of urbanization. This work also provides a reference for the construction of new urbanization in other provinces and cities.

## 2. Data and Methods

### 2.1. Data Sources

The research sample of this study included 11 prefecture-level cities in Zhejiang Province, China, and the research period was 2011–2020. The study was primarily based on the statistical data for social and economic development in Zhejiang Province and the vector map of the municipal administrative divisions in Zhejiang Province. The statistical data primarily came from the *Zhejiang Statistical Yearbook* issued by the Zhejiang Provincial Bureau of Statistics from 2012 to 2021, as well as the statistical yearbook of various cities, the *Zhejiang Construction Statistical Yearbook*, the statistical bulletin of the national economic and social development of Zhejiang Province and cities, and the Statistical Yearbook of the Natural Resources and Environment of Zhejiang Province. The missing data were supplemented by interpolation according to the values of neighboring years.

### 2.2. Methods

(1)Dimensionless processing

Because the evaluation system involved multiple dimensions and the data units of each dimension were different, the dimensionless data needed to be processed. The processed results were all between 0 and 1. To avoid the impact of a 0 value on subsequent calculations, the dimensionless data were uniformly shifted to the right by 0.001 units.

For positive indicators:(1)$$X_{ijk}=\frac{X_{ijk}-\min(X_{j})}{\max\left(X_{j}\right)-\min(X_{j})}$$

For adverse indicators:(2)$$X_{ijk}=\frac{\max(x_{j})-x_{ijk}}{\max(x_{j})-\min(x_{j})}$$
where *i* represents the year (*i* = 2011, 2012, …, 2020), *j* represents various indicators (*j* = 1, 2, …, 22), *k* represents all cities in Zhejiang Province (*k* = 1, 2, …, 11), $\max(x_{j})$ represents the maximum value of the *j*th index, and $\min(x_{j})$ represents the minimum value of the *j*th index. $x_{ijk}$ represents the original data, and $x_{ijk}$ represents the processed data.

(2)Entropy method

The entropy method is used to determine the weight of each index according to the information content of each index. It is an objective evaluation method that avoids the impact of subjective factors on the evaluation object. The smaller the entropy value, the smaller the uncertainty of the data and the smaller the uncertainty of the results with the increasing amount of information. The proportion of the *j*th index to the *k*th city in the *i*th year is calculated as follows:(3)$$p_{ijk}=\frac{X_{ijk}}{\sum_{i=1}^{21}\sum_{k=1}^{11}X_{ijk}}$$

The entropy value of index *j* is calculated as follows:(4)$$e_{j}=-K\sum_{i=1}^{21}\sum_{k=1}^{11}\left(p_{ijk}\ln p_{ijk}\right)$$

The weight of index *j* is calculated as follows:(5)$$a_{j}=\frac{1-e_{j}}{\sum_{j=1}^{22}(1-e_{j})}$$

The composite index is calculated as follows:(6)$$U_{ik}=\sum_{j=1}^{22}a_{j}p_{ijk}$$

In the formula, $U_{\mathrm{ik}}$ represents the comprehensive index of urbanization quality of the city *k* in the *i*th year.

### 2.3. Index System

Based on existing research results, the principles of representativeness, and the objectivity and operability of the indicators, in this study, 22 indicators were designed based on four aspects: the quality of economic development, the quality of the ecological environment, the quality of urban and rural coordination, and the quality of public services. Additionally, the urbanization level of Zhejiang Province was evaluated by integrating the quality of each indicator, as shown in Table 1. The quality of economic development primarily reflects the growth of the regional economy, the adjustment and optimization of the industrial structure, and the degree of investment in innovation and research. The quality of the ecological environment primarily measures the coordination between urbanization and resources and environment. The quality of urban and rural overall planning is primarily used to reflect the strength of regional urban and rural overall planning and the coordination of urban and rural development. The quality of public services primarily reflects the regional infrastructure construction and supply capacity, as well as the quality of life of residents from the supply side. To provide people with more effective public services and solve the problem of insufficient and uneven supply of public services, the key feature of China’s current supply-side reform has become to increase the types and quantity of basic public services, such as education, medical care and old-age care, realize its balanced development, and improve the quality of people’s life.

## 3. Results

### 3.1. Characteristics of Urbanization Quality and Time Series Development

The evaluation results were obtained through the comprehensive evaluation of the urbanization quality of 11 prefectures and cities in Zhejiang Province from 2011 to 2020, as shown in Table 2. The results show that the overall urbanization quality of Zhejiang Province steadily improved during the study period, and the regional gap increased year by year from 2011 to 2015. The regional gap reached its peak in 2015 and narrowed after 2016, becoming relatively stable. The average score of urbanization quality in Zhejiang Province in 2011 was 0.349, and the average score in 2015 was 0.444, with a five-year growth rate of 27.14%. The average score of urbanization quality in 2020 was 0.495, with a growth rate of 41.80% during 2011–2020, indicating that the urbanization quality of Zhejiang Province has basically shown a steady improvement trend. However, it is worth mentioning that, after 2019, with the effects of the pandemic, in 2020, except for Wenzhou and Shaoxing, the indicators of other cities declined by varying degrees, and the quality of urbanization declined. From the perspective of development trends, all cities in the province showed a relatively stable growth trend during 2011–2015. Quzhou, Taizhou, and Lishui continued to grow in 2019. The urbanization quality of the remaining eight cities showed a declining trend in 2016 and increased steadily after 2017. The urbanization quality for Hangzhou, Ningbo, Jiaxing, and Shaoxing in 2017 was lower than that in 2015, as shown in Figure 1.

From the perspective of the development axis of each city, as shown in Figure 2a, Lishui and Quzhou have achieved a relatively rapid cumulative growth rate during the study period, with a growth rate of more than 100%. The quality of urbanization has reached more than twice the original level and is in the first echelon of the development rate of urbanization quality. Additionally, Taizhou City, Wenzhou City, and Shaoxing City are in the second tier of urbanization quality development rate. Their cumulative growth rate during the study period is higher than the average level of the province. In addition to Zhoushan City, which has a slightly lower growth rate than the provincial average, the urbanization growth rate of the other five cities is far lower than the provincial average.

During 2011–2015, as shown in Figure 2b, the average growth rate of urbanization quality in the whole province was 27.13%. Shaoxing City had the largest growth rate of 43.00%, followed by Taizhou City, Wenzhou City, Quzhou City, and Lishui City, with growth rates of 41.83%, 39.79%, 38.75%, and 37.00%, respectively. These were in the first echelon of the growth rates. The growth rates of Zhoushan City and Jiaxing City were 28.01% and 25.11%, respectively, close to the average level of the whole province. These cities were in the second echelon of the growth rates. Huzhou City had the smallest growth rate, only 11.00%, and Jinhua City had the second smallest growth rate, 15.85%, which was in the fourth echelon of the growth rates. The growth rates of the other cities in Hangzhou and Ningbo were also small, 20.39% and 21.98%, respectively, which were in the third tier of the growth rates.

During 2016–2020, as shown in Figure 2c, the average growth rate of urbanization quality in the province was 19.60%. Jiaxing City and Ningbo City ranked first in terms of the growth rate of urbanization, with growth rates of 30.11% and 30.02%, respectively. Huzhou City, Shaoxing City, and Zhoushan City ranked second in terms of the growth rate of urbanization, with growth rates of 24.24%, 22.02%, and 21.42%, respectively. The growth rates of urbanization quality in Jinhua City and Wenzhou City were at the average level of the province, at 20.52% and 19.05%, respectively, which were in the third echelon of urbanization quality growth. The growth rates of the urbanization quality in Quzhou, Taizhou, Lishui, and Hangzhou were lower than the average level of the province, at 14.74%, 13.34%, 11.61%, and 12.00%, respectively. These were in the fourth echelon of urbanization quality growth.

### 3.2. Spatial Difference of Urbanization Quality

The spatial distribution of urbanization quality in Zhejiang Province is uneven, as shown in Figure 3. Hangzhou and Ningbo have been in the first echelon of the province’s urbanization quality development across the study period. Jiaxing’s urbanization quality was ranked third in the province until 2015. After 2016, Zhoushan City replaced Jiaxing City to take third place in the province with respect to urbanization quality, and the urbanization quality of Zhoushan City has been higher than the average level of the province for the study period. The urbanization qualities of Quzhou City, Lishui City, Jinhua City, Taizhou City, and Wenzhou City were lower than the average level of the province. In 2011 and 2012, the urbanization quality of Huzhou City was higher than the provincial average level and then lower than the provincial average level across the next 8 years. Jiaxing’s urbanization quality in the 3 years from 2016 to 2018 was lower than the provincial average, and the urbanization quality across the remaining 7 years was higher than the provincial average. In 2015–2017 and 2020, the urbanization quality of Shaoxing was higher than the provincial average, and the urbanization quality across the remaining 6 years was lower than the provincial average. The extreme difference and standard difference of urbanization quality in 11 cities in Zhejiang Province showed that the spatial difference of urbanization quality development in each city first expanded and then shrank. The regional differences increased from 2011 to 2015, reaching a peak in 2015, and gradually narrowed after 2016.

According to the comprehensive evaluation value of urbanization quality accumulated during 2011–2020, Hangzhou has the highest cumulative comprehensive value of 6.880, followed by Ningbo, at 5.411, and Zhoushan and Jiaxing, at 4.771 and 4.606, respectively. The evaluation value of these four cities was higher than the average cumulative value of 4.328, and the evaluation value of the other seven cities was lower than the average value. The last two cities were Lishui and Quzhou, with values of 3.327 and 3.289, respectively, less than half of that of Hangzhou. This shows that the urbanization quality of most cities in Zhejiang Province still has great room and potential for improvement.

In 2020, as shown in Figure 3a, the quality of urbanization was high in the north, lower in the south, and lowest in the middle. Hangzhou had the highest urbanization quality, reaching 0.731, which was 1.48 times the provincial level. The urbanization quality of Ningbo City, Zhoushan City, Shaoxing City, and Jiaxing City were higher than the provincial average level, at 0.577, 0.543, 0.504, and 0.500, respectively. The urbanization quality of Jinhua City was the lowest, at 0.392, and that of the rest of the cities–Huzhou City, Wenzhou City, Taizhou City, Lishui City, and Quzhou City–was lower than the provincial average level, at 0.460, 0.445, 0.439, 0.430, and 0.426, respectively.

In 2011, as shown in Figure 3b, the spatial level difference in the urbanization quality was obvious, and the overall pattern was high in the northeast and the south coast and low in the southwest and in the inland. The urbanization quality of Hangzhou was 0.602, 1.72 times the average level of the whole province. This was far greater, and 0.112 higher, than that of Ningbo in second place. However, Ningbo was also significantly ahead of other cities (0.087 higher than that of Jiaxing in third place). The urbanization quality values for Jiaxing City, Zhoushan City, and Huzhou City were 0.402, 0.392, and 0.373, respectively, higher than the provincial average level. The urbanization quality values for Lishui City and Quzhou City were the lowest, at 0.214 and 0.213, respectively; these were only half of the provincial average level and only 1/3 of the value for Hangzhou City, which had the highest urbanization quality. The urbanization quality values for Jinhua City, Shaoxing City, Wenzhou City, and Taizhou City were also lower than the provincial average, at 0.319, 0.311, 0.271, and 0.255, respectively.

In 2015, as shown in Figure 3c, the quality of urbanization remained high in the northeast and low in the southwest. The urbanization quality for Hangzhou still ranked first, reaching 0.725, 1.63 times the provincial average and 0.128 times higher than that of Ningbo, in second place. The urbanization quality values for Jiaxing and Zhoushan were higher than 0.5 (0.503 and 0.502, respectively). The urbanization quality value for Shaoxing was also higher than the provincial average, at 0.445, higher than the provincial average for the first time since 2011, and then essentially equal to or slightly lower than the provincial average. The urbanization quality values for Lishui City and Quzhou City were still the lowest, at 0.293 and 0.296, respectively, but the relative gap with the provincial average narrowed. The ranking of Lishui City and Quzhou City has improved since 2016, and it is no longer the lowest in the province. This indicates that the urbanization quality development for Lishui City and Quzhou City has begun to catch up with each other. The concept of “green water and green mountains are golden water and silver mountains” has been well implemented and reflected. In addition, Huzhou City, Wenzhou City, Jinhua City, and Taizhou City have urbanization quality values of 0.414, 0.379, 0.369, and 0.361, respectively. These values were lower than the provincial average. Jinhua City has had the lowest urbanization quality in the province since 2016, except in 2019, when it was slightly higher than Wenzhou City, which was at the bottom. This shows that Jinhua lacks the stamina and impetus for urbanization development. The urbanization quality of Wenzhou has also been relatively low in the province, and its development potential needs to be explored and broken through.

### 3.3. Analysis of the Correlation between Urbanization Quality and Evaluation Indicators

#### 3.3.1. Gray Correlation Calculation Method

Gray correlation is a method that can be used for systematic analysis to determine the impact of each index factor on the final result. The basic idea of Gray Relation Analysis (GRA) is to judge whether the relationship is tight according to the similarity of the shape of the series curve set. The closer the curve, the greater the degree of correlation between related orders, and vice versa [38].

Gray correlation analysis is a research method that measures the degree of correlation between data by studying the degree of correlation between data (the degree of correlation between the parent series and feature series) and by measuring the degree of correlation between the data through the degree of correlation, thus assisting decision-making.

The value range of the gray correlation degree is [0, 1]. The larger the value, the stronger the correlation between the gray correlation degree and the parent sequence.

#### 3.3.2. Gray Relational Degree Result Analysis

Using the gray correlation analysis method, the correlation degree between the quality of urbanization and each evaluation index was obtained.

Among the top 10 evaluation indicators, as shown in Table 3, the quality of urbanization has the highest correlation with the reduction rate of energy consumption per unit gross domestic product. From the perspective of energy consumption, this shows that the improvement of resource utilization efficiency and the green transformation of economic intensive development are of great significance to the high-quality development of urbanization. The degree of correlation with the proportion of days with good air quality is also ranked seventh, showing that the improvement of the ecological environment quality is conducive to meeting the people’s growing need for a beautiful ecological environment; this can promote the realization of higher quality, more efficient, and more sustainable development. The Engel coefficient ratio of urban and rural residents, the per capita disposable income ratio of urban and rural residents, and the per capita consumption expenditure ratio of urban and rural residents are ranked from second to fourth. These values are typical representatives of the quality of urban-rural integration development and can reflect the coordinated development of urban and rural areas. This fully demonstrates that the integration of urban and rural development, the narrowing of the development gap between urban and rural areas, the gradual changing of the dual economic structure of urban and rural areas, and the promotion of coordinated urban and rural development are some of the important manifestations of high-quality urbanization development. The fifth evaluation index reflects the close relationship between the proportion of the output value of the tertiary industry and the quality of urbanization, indicating that the tertiary industry is the main driving force of modern urbanization, and that it is necessary to reasonably promote the transformation of the economic development mode and the adjustment of the economic structure and further promote the tertiary industry. Industrial development will further promote the development of urbanization quality. The proportion of the urban population to the total population, as the sixth evaluation indicator, reflects not only the proportion of the urban population to the total population but also the scale and rate of migration from the rural population to the urban population. As China’s urbanization has risen above 50%, improving the quality of urbanization has become a vital and important task of urbanization. However, this also shows that the level of urbanization or the increase in the proportion of the urban-rural population still has a basic position. To ensure the quality of urbanization, it is necessary to continue to maintain a rapid development rate. The eighth and ninth evaluation indicators reflect the fact that the level of public services, such as education and health, also has a significant impact on the quality of urbanization. The tenth evaluation indicator, the average wage of on-the-job employees in urban units above the designated size, shows that the level of wage income of employees in a region also plays an important role in the quality of urbanization. Overall, the urbanization development in Zhejiang Province has broken through the limitations of traditional urbanization. The quality of urbanization is closely related to the ecological environment and the overall development of urban and rural areas, as well as to economic development and public service levels. However, the impact of innovation and research and development level on the quality of urbanization is still relatively weak, and it is at the bottom of the correlation ranking. This requires Zhejiang Province to pay more attention to the quality of the ecological environment, the integration of urban and rural areas, and the improvement of public services in future urbanization development. Moreover, this encourages innovative research and development, gives full play to its spillover effect, and promotes the improvement of the quality of new urbanization.

#### 3.3.3. Suggestions

According to the research results, the quality of urbanization in Zhejiang Province varies and there is an imbalance related to various factors such as the geographical location, economic base, public services, ecological environment, and related policies of various cities. In the context of regional coordinated development, relying on the opportunity of the construction of a common prosperity pilot demonstration area, the following four reference suggestions are put forward for improving the quality of new urbanization in Zhejiang Province:(1)Investment in innovation should be increased, employment support for urban industries should be strengthened, and the coordinated development of urban and rural areas should be promoted. In the process of economic development, Zhejiang Province should pay attention to the optimization and upgrading of industrial infrastructure, increase investment in innovative research and development, properly maintain the development of the secondary industry, and focus on increasing the proportion of the tertiary industry. For example, Wenzhou has been at a low level of urbanization quality during the study period. The fundamental reason for this is that Wenzhou is a city that was developed by relying on the private economy and township enterprises. The urban–rural dual structure has not been completely changed and the transformation of industrial structure has lagged, thus lowering the urbanization quality level.(2)Geographical advantages should be taken advantage of, urban public service capacity should be improved, and high-quality smart cities should play a leading role. Areas with high urbanization quality, such as Hangzhou and Ningbo, should continue to give full play to their advantages, take advantage of the superior geographical location to actively connect the construction of the Yangtze River Delta urban agglomeration and the national strategic development, improve the comprehensive carrying capacity and service capacity of the city and give full play to its radiating and driving role in the surrounding areas, promote coordinated development with the surrounding areas, build a metropolitan area, pay attention to the coordinated development between the economy and the ecological environment, improve the quality of public services, and build a leading area for China’s economic transformation, upgrading, reform, and innovation.(3)Attention should be paid to the inland mountainous areas, the coordinated development of the whole region should be promoted, and the mountain and sea cooperation project should be strengthened. The southwest area of Zhejiang Province has always been an underdeveloped region in the province, with poor geographical conditions and a weak urbanization foundation, resulting in a low level of urbanization. The Zhejiang Provincial Government has promoted the “Mountain and Sea Cooperation Project” for more than 20 years, and its policy effect is obvious. This is reflected in the prominent growth of the urbanization quality in Quzhou and Lishui, which are included in the project and have entered the rising channel. However, there are also policy marginal areas similar to Jinhua, and the urbanization development momentum is insufficient. It is necessary to pay attention to this over time, adjust the policy coverage of the mountain and sea cooperation project, and effectively promote coordinated development in the region.(4)Environmental governance should be continued, green and low-carbon development should be encouraged, and people’s quality of life should be improved. The direction of new urbanization toward sustainable development, that is resource-saving, environmentally friendly, economically efficient, and socially harmonious and that constantly improves the quality of people’s lives, is increasingly clear. The impact of the ecological environment on the quality of urbanization is being increasingly weighed. Zhejiang Province should unswervingly integrate the concept of ecological civilization into the process of urbanization, adhere to the path of intensive, efficient, green, and low-carbon sustainable development, and pay attention to the effectiveness of comprehensive environmental governance. Regions with a good ecological environment should maintain the advantages of the ecological environment, effectively improve the ecological function, actively and steadily improve their economic development level, and promote coordinated development within the region, passing on the principle of not damaging the ecological environment.

## 4. Discussion and Conclusions

The results of the comprehensive measurement of urbanization quality indicated that the overall urbanization quality of Zhejiang Province showed a steady improvement trend, with the growth rate in the first five years (2015–2011) being higher than that in the subsequent five years (2016–2020). From the perspective of regional differences, there is a large gap in the quality of urbanization development between cities. The regional differences increased from 2011 to 2015, reached a peak in 2015, and gradually narrowed after 2016. Hangzhou has the highest urbanization quality, with a 10-year cumulative measured value of 6.88, and has always been in the first place, followed by Ningbo, which is ahead of other cities but has a certain gap with Hangzhou. The order of urbanization quality development in other cities is in a fluctuating state. Quzhou has the lowest cumulative measured value over the 10-year period, with an urbanization quality development index of 3.29, followed by Lishui, with a value of only 3.33, less than half of that of Hangzhou. In terms of spatial distribution, the overall pattern of “high in the northeast and low in the southwest, high in the south and low in the middle” in 2011 evolved into a pattern of “medium in the north and south, high in the north and low in the south and low in the middle” in 2020. By comparison, the urbanization quality of the southwest inland mountainous areas has developed rapidly and improved significantly, while the urbanization quality development level of Huzhou, which is at the north end of the province, has declined significantly. Through the analysis of the correlation degree, it was found that the development of urbanization in Zhejiang Province broke through the limitations of traditional urbanization. The quality of urbanization is closely related to the ecological environment and the overall development of urban and rural areas, as well as the economic development and public service level. However, the impact of innovation, research, and the development level on the quality of urbanization is still weak and is at the bottom of the ranking of the correlation degree. This requires Zhejiang Province to pay more attention to the ecological environment quality, urban-rural integration, and the improvement of the public service level while encouraging innovative research and development, giving full play to its spillover effect and promoting the quality of new urbanization. Specific suggestions for this are as follows: (1) Increase investment in innovation, strengthen support for urban industrial employment, and promote the coordinated development of urban and rural areas. (2) Take advantage of geographical advantages to improve urban public service capacity and play a leading role. (3) Pay attention to the inland mountainous areas, promote the coordinated development of the whole region, and strengthen the mountain and sea cooperation project. (4) Continue environmental governance, adhere to green and low-carbon development, and improve people’s quality of life. In the process of urbanization construction, all cities in Zhejiang Province should give full play to their advantages and should compensate for their weaknesses, which will inevitably narrow the regional differences between regions and improve the overall urbanization quality development level of Zhejiang Province.

The study use the entropy weight method to quantitatively measure the urbanization quality of 11 cities in Zhejiang Province. The entropy weight method is more accurate and objective than the subjective evaluation method and could better explain the results obtained. However, its limitation lies in that it only depends on the volatility of the data, or the so-called amount of information, to obtain the weight without considering the actual significance of the data. Lacking the analysis of how the factors interact with each other may lead to a deviation in the results. As such, the gray correlation analysis method is used in this study to analyze the correlation degree between the quality of urbanization and each evaluation index. The follow-up study will use different research methods for verification.

## Figures and Tables

**Figure 1 ijerph-20-04093-f001:**
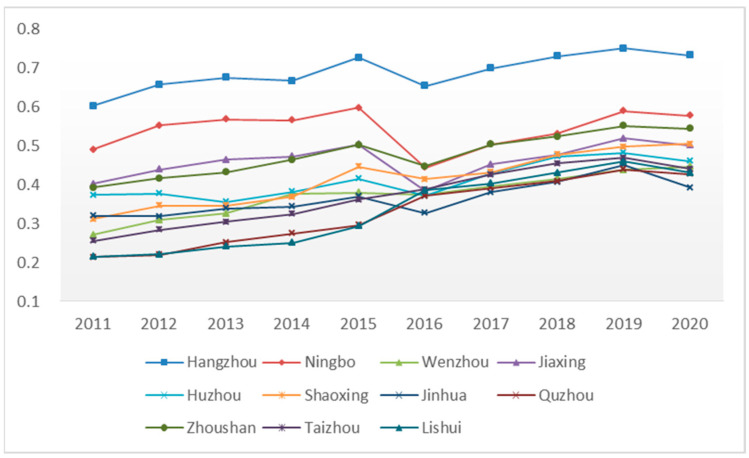
Schematic diagram of changes in urbanization quality of 11 cities in Zhejiang Province.

**Figure 2 ijerph-20-04093-f002:**
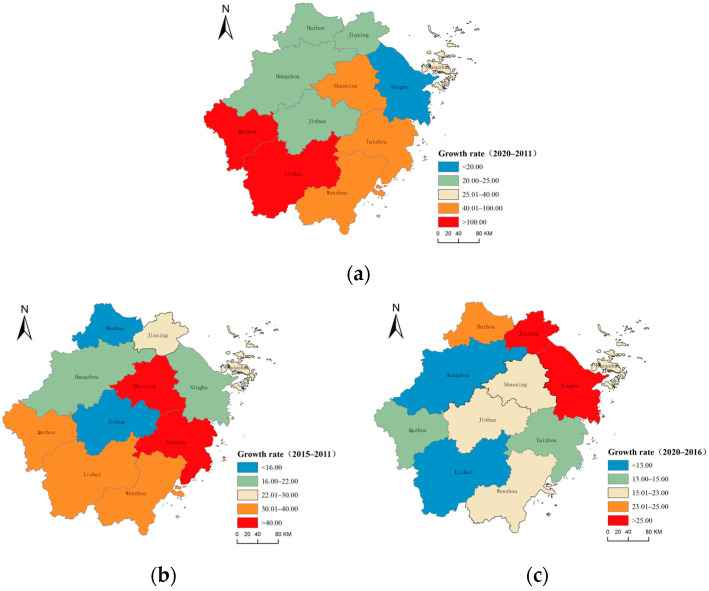
Spatial distribution of urbanization quality growth in Zhejiang Province: (**a**) 2020–2011; (**b**) 2015–2011; (**c**) 2020–2016.

**Figure 3 ijerph-20-04093-f003:**
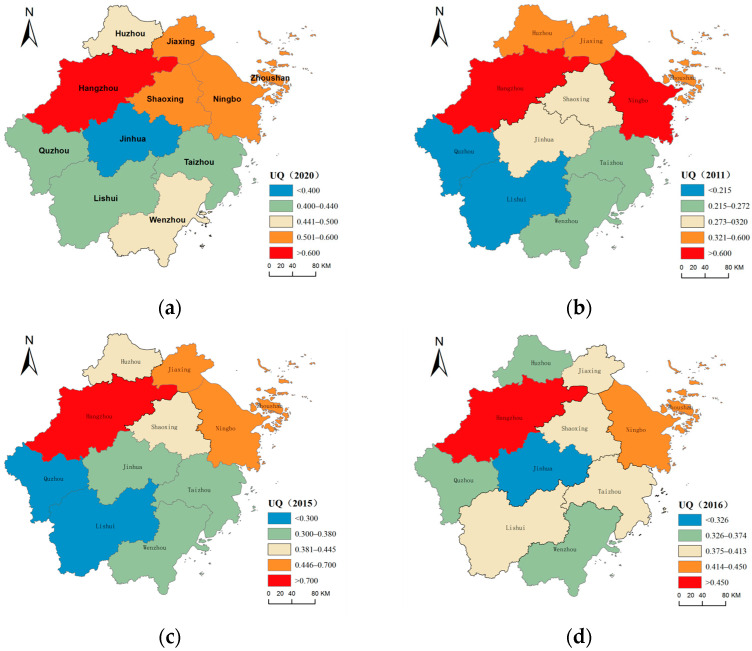
Spatial distribution of urbanization quality in Zhejiang Province: (**a**) 2020; (**b**) 2011; (**c**) 2015; (**d**) 2016.

**Table 1 ijerph-20-04093-t001:** Urbanization quality evaluation index system and weight.

First-Level Index	Second-Level Index	Unit	Index Type	Weight
Economic development quality	Per capita gross domestic product (x1)	RMB	Positive	0.060754
	Per capita local fiscal revenue (x2)	RMB	Positive	0.091792
	Proportion of tertiary industry output value (x3)	%	Positive	0.066676
	Proportion of R and D expenditure in gross domestic product (x4)	%	Positive	0.041799
	Number of patents authorized per 10,000 people (x5)	Piece	Positive	0.072680
Ecological environment quality	Per capita park green area (x6)	m^2^	Positive	0.018217
	Reduction rate of energy consumption per unit gross domestic product (x7)	Ton standard coal/10^4^ RMB	Positive	0.003321
	Greening coverage rate of built-up area (x8)	%	Positive	0.010525
	Proportion of days with good air quality (x9)	%	Positive	0.033225
	Average annual concentration of fine particles (PM10) (x10)	%	Negative	0.003679
Quality of urban and rural overall planning	Proportion of the urban population to the total population (x11)	%	Positive	0.035892
	Per capita disposable income ratio of urban and rural residents (x12)	%	Negative	0.015361
	Per capita consumption expenditure ratio of urban and rural residents (x13)	%	Negative	0.014341
	Per capita disposable income growth rate of urban residents (x14)	%	Positive	0.045245
	Engel coefficient ratio of urban and rural residents (x15)	%	Positive	0.058120
	Average wage of on-the-job employees in urban units above the designated size (x16)	RMB	Positive	0.061287
Public service quality	Number of students in ordinary primary and secondary schools per 10,000 people (x17)	Per 10^4^ person	Positive	0.042722
	Number of health technicians per 10,000 people (x18)	Per 10^4^ person	Positive	0.055554
	Number of beds in health institutions per 10,000 people (x19)	Per 10^4^ person	Positive	0.051835
	Book collection in public library per 10,000 people (x20)	Per 10^4^ person	Positive	0.090099
	Number of public transport options per 10,000 people (x21)	Per 10^4^ person	Positive	0.067877
	Urban road area per capita (x22)	m^2^	Positive	0.059000

**Table 2 ijerph-20-04093-t002:** Comprehensive index of urbanization quality in Zhejiang Province from 2011 to 2020.

	2011	2012	2013	2014	2015	2016	2017	2018	2019	2020
Hangzhou	0.602	0.656	0.674	0.666	0.725	0.653	0.698	0.729	0.749	0.731
Ningbo	0.489	0.551	0.567	0.565	0.597	0.443	0.503	0.530	0.588	0.577
Wenzhou	0.271	0.308	0.325	0.375	0.379	0.374	0.396	0.415	0.437	0.445
Jiaxing	0.402	0.438	0.463	0.472	0.503	0.384	0.451	0.475	0.518	0.500
Huzhou	0.373	0.377	0.355	0.381	0.414	0.371	0.428	0.472	0.481	0.460
Shaoxing	0.311	0.345	0.346	0.368	0.445	0.413	0.431	0.476	0.497	0.504
Jinhua	0.319	0.318	0.338	0.343	0.369	0.326	0.380	0.406	0.450	0.392
Quzhou	0.213	0.219	0.251	0.274	0.296	0.371	0.391	0.409	0.438	0.426
Zhoushan	0.392	0.416	0.431	0.463	0.502	0.447	0.503	0.523	0.550	0.543
Taizhou	0.255	0.283	0.304	0.324	0.361	0.387	0.425	0.454	0.468	0.439
Lishui	0.214	0.221	0.241	0.250	0.293	0.385	0.403	0.430	0.459	0.430
Total	3.841	4.132	4.296	4.483	4.884	4.554	5.009	5.320	5.636	5.447

**Table 3 ijerph-20-04093-t003:** Top 10 indicators for the correlation between urbanization quality and 22 indicators.

Ranking	Evaluation Index	Correlation
1	Reduction rate of energy consumption per unit gross domestic product (x7)	0.9884
2	Engel coefficient ratio of urban and rural residents (x15)	0.9708
3	Per capita disposable income ratio of urban and rural residents (x12)	0.9682
4	Per capita consumption expenditure ratio of urban and rural residents (x13)	0.9666
5	Proportion of tertiary industry output value (x3)	0.9649
6	Proportion of the urban population to the total population (x11)	0.9586
7	Proportion of days with good air quality (x9)	0.9578
8	Number of students in ordinary primary and secondary schools per 10,000 people (x17)	0.9517
9	Number of health technicians per 10,000 people (x18)	0.9461
10	Average wage of on-the-job employees in urban units above the designated size (x16)	0.9428

## Data Availability

The data supporting the findings of this study are available within the article.

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
