# Peer review of "Analysis of the Spatial-Temporal Evolution of Urbanization Quality in Zhejiang Province, China"

_ijerph, 2023, doi:10.3390/ijerph20054093_

Round 1
Reviewer 1 Report
Data sources are not clearly stated. How reliable are they?
Describe please, the method of data aggregation and comparison, in order to achieve the comparative figures of each measurement. There is no explanation of
Describe, what is meant under the expression “quality of life” in the human perspective in China? There is a huge gap in understanding of this expression in different parts of the World.
The references are dominated by the Chinese authors. You should compare your findings with more books and articles coming from other countries in order to build the “World Realm of Knowledge” on urbanisation and development of life quality of the inhabitants life. Social, cultural and educational factors should be examined also in the further scientific investigations.
More graphical schemes should be attached to improve the understanding of the study, like urban growth schemes of selected cities in the following years of the time span analysed, photos describing the physical examples of positive developments, etc.
Involvement of the individuals in the article should be evidenced and described.
Reviewer 2 Report
Thank you for giving me this opportunity to read the manuscript entitled " Analysis of the Spatial-Temporal Evolution of Urbanization Quality in Zhejiang Province, China". The study provides a comprehensive analysis of the urbanization quality of cities in Zhejiang Province and identifies the factors that influence the quality of urbanization. The results of the study can be useful for local governments in formulating feasible urbanization development strategies and policies to promote high-quality development of urbanization. However, the study could be improved by providing more details on the methodology used to construct the evaluation index system and a more comprehensive analysis of the impact of other factors on the quality of urbanization.
1. Please replace the keywords that already appear in the manuscript's title with close synonyms or other keywords, which will also facilitate your paper being searched by potential readers.
2. More discussion regarding the potential limitations and biases of the entropy weight method used for measuring urbanization quality should be provided.
3. The paper does not provide a comprehensive analysis of the impact of factors such as population growth, migration, and urban-rural integration on the quality of urbanization.
4. The study only used the entropy weight method to measure the urbanization quality of the 11 cities in Zhejiang Province. Other evaluation methods could have been used to verify the results.
5. Line 33, “… become the goal of China’s new urbanization development, …”, a paper titled “How does urban expansion impact people’s exposure to green environments? A comparative study of 290 Chinese cities” is suggested to be citied as references to support the statement here.
6. The paper discusses the factors that influence the quality of urbanization in Zhejiang Province, however, it does not provide an in-depth analysis of how these factors interact with each other. A more detailed examination of the relationships between these factors could enhance the understanding of the urbanization process and improve the quality of the analysis.
